# The Comparative Genomics of Botryosphaeriaceae Suggests Gene Families of *Botryosphaeria dothidea* Related to Pathogenicity on Chinese Hickory Tree

**DOI:** 10.3390/jof10040299

**Published:** 2024-04-22

**Authors:** Dong Liang, Yiru Jiang, Yu Zhang, Chengxing Mao, Tianlin Ma, Chuanqing Zhang

**Affiliations:** College of Advanced Agricultural Sciences, Zhejiang Agriculture and Forest University, Hangzhou 311300, China; liangdong@zafu.edu.cn (D.L.); 19906839001@163.com (Y.J.); zy1335659339@163.com (Y.Z.); zafumaocx@163.com (C.M.)

**Keywords:** Chinese hickory, Botryosphaeriaceae, high-quality genome assemblies, pathogenicity, colonization

## Abstract

Trunk canker poses a major threat to the production of Chinese hickory tree (*Carya cathayensis* Sarg.), which is primarily determined by Botryosphaeriaceae. In our previous work, we identified *Botryosphaeria dothidea* as the predominant pathogen of this disease. However, it is still unclear about corresponding gene families and mechanisms associated with *B. dothidea*’s pathogenicity on Chinese hickory tree. Here, we present a comparative analysis of high-quality genome assemblies of *Botryosphaeria dothidea* and other isolated pathogens, showing highly syntenic relationships between *B. dothidea* and its closely related species and the conservative evolution of the Botryosphaeriaceae family. Higher GC contents were found in the genomes of *B. dothidea* and three other isolated pathogens (*Botryshaeria cortices*, *Botryshaeria fabicerciana,* and *Botryshaeria qingyuanensis*) compared to *Macrophomina phaseolina*, *Neofusicoccum parvum*, *Diplodia corticola,* and *Lasiodiplodia theobromae*. An investigation of genes specific to or expanded in *B. dothidea* revealed that one secreted glucanase, one orsellinic acid biosynthesis enzyme, and two MFS transporters positively regulated *B. dothidea*’s pathogenicity. We also observed an overrepresentation of viral integrase like gene and heterokaryon incompatibility proteins in the *B. dothidea*’s genome. In addition, we observed one LRR-domain-containing protein and two Sec-domain-containing proteins (Sec_1 and Sec_7) that underwent positive selection. This study will help to understand *B. dothidea*’s pathogenicity and potential influence on the infection of Chinese hickory, which will help in the development of disease control and ensure the security of Chinese hickory production.

## 1. Introduction

The Botryosphaeriaceae family are worldwide pathogens that cause a range of disease symptoms, including leaf spots, fruit and root rots, dieback, and cankers, in multiple woody hosts [1,2,3]. They have diverse ecological roles as endophytic, saprobic, and plant pathogenic species [4]. Botryosphaeria is the largest genus of Botryosphaeriaceae, with *Botryosphaeria dothidea* being a typical species, followed by other genera including Diplodia, Dothiorella/Spencermartinsia, Lasiodiplodia, and Neofusicoccum [5]. The symbolic life cycle of *B. dothidea* is characterized by an endophytic/latent infection phase [4,6]. It exists as an endophyte within the host plant, and its virulence is triggered by the host plant’s physiological status in response to environmental stressors. This stimulation leads *B. dothidea* to transition into a pathogenic state. Thus, the manifestation of symptoms in *B. dothidea* infections appears to be influenced by the occurrence of a single environmental stress event or a combination of environmental stress events [7,8]. This fungus can parasitize living host plants and gains infection status under the stimulation of host or environmental stress, which results in obvious disease symptoms [9]. In addition to *B. dothidea*, its closely related species, such as *B. cortices*, *B. fabicerciana*, and *B. qingyuanensis*, were also isolated from multiple diseased host plants (like *Malania oleifera*, *Mangifera indica,* and so on) [10,11]. Moreover, *M. phaseolina*, which belongs to the Macrophomina genus of Botryosphaeriaceae, is a globally distributed soil-borne fungus that causes diseases like dry root rot and charcoal rot in various crops, like soybean and maize [12,13]. *N. parvum* is a representative species of the Neofusicoccum genus of Botryosphaeriaceae and with worldwide distribution. This fungus has been associated with dieback diseases in various plant species, including grapevines, citrus, and almond trees, highlighting its significant economic impact on agriculture [14,15]. *D. corticola*, an emerging canker pathogen from the Diplodia genus, has garnered attention due to its association with oak species in various regions, including the United States and Europe [16]. *L. theobromae* has a wide geographic distribution as the biotic agent that induces copious necrosis and gummosis, eventually resulting in the reduced vigor and lifespan of many economically important woody trees, including cacao [17] and citrus [18].

Chinese hickory (*Carya cathayensis* Sarg.), a woody oil species belonging to the Juglandaceae family, has gained significant economic importance due to its production of edible nuts. It is primarily distributed in the Lin’an district located in southeastern China [19]. However, trunk canker disease caused by *B. dothidea* continuously threatens Chinese hickory production in this region and has already resulted in significant economic losses. The pathogenic status of *B. dothidea* is activated by optimal temperatures (around 25 °C) during this process [20]. In our previous work, colleagues collected samples of trunk canker disease, encompassing 60% of the Chinese hickory tree yield in the Lin’an district [21]. Five Botryosphaeriaceae species (*B. dothidea, B. qingyuanensis, B. cortices, B. fabicerciana, and Lasiodiplodia theobromae*) were isolated, and 89 of 96 isolates were identified as *B. dothidea*. This revealed the dominant role of *B. dothidea* in trunk canker disease samples in the Lin’an district, which might be related to its stronger colonization capacity or pathogenicity. The associated mechanisms underlying this phenomenon remain unclear. Through comparative genomics analysis, it will help us to unveil the evolutionary process of Botryosphaeriaceae. More importantly, investigation of *B. dothidea* gene families associated with its pathogenicity will promote understanding of the underlying molecular mechanism.

Although multiple sequenced genomes of *B. dothidea* were proposed [22], our lab first proposed high-quality genome assemblies of *B. dothidea* coupled with other species mentioned above [23]. Based on this work, we further present a comprehensive comparative genome analysis of these species with the following objectives: (i) to investigate *B. dothidea* pathogenesis-related genes that participate in the infection of Chinese hickory, especially candidates associated with the formation of latent infection, and (ii) to explore specific genes that might indicate *B. dothidea*’s advantage within the community inside trunk canker disease of Chinese hickory.

The current control strategies for hickory trunk canker disease mainly rely on physical or chemical measures, such as scraping off the canker lesions on the trunk surface. However, they are confronted with issues of poor efficiency, which result from a lack of in-depth understanding of the molecular mechanisms behind *B. dothidea*’s pathogenicity in Chinese hickory. Through comparative genomics analysis, this study reveals pathogenicity-related genes that are specific to or expanded in *B. dothidea*. It provides clues for the deepening understanding of *B. dothidea*’s pathogenic mechanisms in hickory and may also aid in the design of fungicides targeting these pathogenicity-related genes, thereby improving the prevention and control efficacy of hickory trunk canker disease.

## 2. Materials and Methods

### 2.1. Fungal Growth Conditions, DNA Isolation, and Genomic Sequencing

*B. dothidea* strain BDLD16-7 (BDLA), *B. cortices* strain 16-35 (BCTK), *B. fabicerciana* strain 18-2 (BFLG), and *B. qingyuanensis* strain 16-30 (BQTK) were originally isolated from trunk canker samples of Chinese hickory trees in Lin’an, Zhejiang province, China. All strains were cultivated on potato dextrose agar (PDA) at 25 °C, and 10-day-old mycelium was harvested. Genomic DNA and messenger RNA were extracted using a genome DNA kit (FastPure Microbiome DNA Isolation Kit, Vazyme Biotech Co., Ltd, Nanjing, China) and an Oligotex-dT mRNA kit (Qiagen Inc., Valencia, CA, USA), respectively. The genomic DNA and mRNA were further purified and sequenced using Oxford Nanopore Technology (ONT) and Illumina HiSeq4000 platforms owned by Biomarker Technologies (www.biomarker.com.cn) (Beijing, China), respectively. Apart from the aforementioned species, genome assemblies and corresponding annotations of *Macrophomina phaseolina*, *Neofusicoccum parvum*, *Diplodia corticola*, and *Lasiodiplodia theobromae* were retrieved from the NCBI or JGI genome websites.

### 2.2. Genome Assembly, Gene Prediction, and Functional Annotation

The genome assemblies of BDLA, BQTK, BCTK, and BFLG had been completed in our previous work [23]. Here, we evaluated the completeness of the genome assemblies using BUSCO v5.12 at the Ascomycota level. RepeatMasker v4.1.2 and RepeatModeler v2.0.1 were employed to detect, categorize, and mask repeats in the genome assemblies. The BRAKER2 toolkit v2.1.5 was used to identify a gene model based on repeat-masked genome assemblies, coupled with the support of external protein (fungi_odb10) and transcriptome data evidence [24]. eggNOG v5 [25] was used to perform the functional annotation of coding genes with an *E*-value threshold of 1 × 10^−5^.

### 2.3. Orthologs Identification and Species Phylogenetic Tree Construction

Ortholog construction was conducted using OrthoFinder v2.5.5 [26] with an *E*-value threshold of 1 × 10^−5^ and an MCL inflation index of 2.0. Multiple alignments of single-copy orthologs were conducted using MAFFT v7.490, and a maximum-likelihood tree of the species in this study was constructed using IQ-TREE v2.0.7 with the default parameters using the auto-detect model setting.

### 2.4. Identification of Rapidly Evolving Ortholog Families

Based on the ortholog families identified using OrthoFinder, CAFE (Computational Analysis of Gene Family Evolution, version 5.0) was used to compare the sizes of the ortholog families of each species to their respective most recent common ancestor (MRCA) using fossil data for species divergence times obtained from the Timetree website (https://timetree.org/, accessed on 8 October 2023). Rapidly evolving ortholog families were estimated with the best-fit global birth/death parameter (*λ*) set at 0.0001 and a *p*-value threshold of 0.01. The ultra-metric tree of each ortholog family was generated using r8s v1.81.

### 2.5. Positive Selection Analysis

Protein codon alignments for each single-copy ortholog were generated using pal2nal v14, and TrimAl v1.4 was applied to remove gaps within the codon alignments. dN/dS ratios (ω) were calculated using the CodeML program of PAML v4.9e utilizing the branch model (model = 2 and NSsites = 0). To pinpoint which *B. dothidea* genes underwent positive selection pressure, branches represented by BDLA genes were designated as foreground branches. Likelihood ratio tests (LRTs) were employed to compare the null hypothesis (fix_omega = 1 and omega = 1) with the alternative hypothesis (fix_omega = 0 and omega = 0.7). The significance of LRT statistics was determined using a χ^2^ distribution. The BDLA genes that underwent positive selection pressure were defined by false discovery rate (FDR)-corrected *p*-values of <0.05 and dN/dS ratios (ω) higher than 1.

### 2.6. Detection of Horizontal Gene Transfer (HGT)

Genome-wide predictions of horizontal gene transfer (HGT) events in BDLA were conducted using the AvP toolkit (https://github.com/GDKO/AvP, accessed on 30 October 2023) [27], which performed BLASTp searches against the Uniref100 database with an *E*-value threshold of 1 × 10^−5^. Subsequently, the hits for each BDLA gene were compiled and categorized based on homologous similarity and the corresponding taxon, which was used for the calculation of the Alien Index (AI) value. The software then utilized IQ-TREE (with the default parameters) to construct phylogenetic trees, identifying candidates similar to hits from distant taxa that might have been acquired through horizontal gene transfer (HGT) events.

### 2.7. Synteny Analysis

DIAMOND v0.8.25 was employed for all vs. all BLASTp searches to detect homologous gene pairs between two corresponding genomes (with an E-value threshold of 1 × 10^−5^ and a -max-target-seqs value of 5). Subsequently, MCScanX was utilized to explore intra- or inter-genomic collinearity, and genome annotations (density of protein-coding genes, LS genes, and repeats) were also input. The visualization of the synteny results was conducted using circus v0.69.

### 2.8. Prediction of Secretome and Small Cysteine-Rich Proteins (SCPs)

SignalP v5.0 (https://services.healthtech.dtu.dk/services/SignalP-5.0/, accessed on 28 September 2023) was used to identify signal peptides, and the transmembrane domains of each protein were identified using TMHMM v2.0 (https://services.healthtech.dtu.dk/services/TMHMM-2.0/, accessed on 26 September 2023). EffectorP v3.0 (https://effectorp.csiro.au/, accessed on 29 September 2023) was used to predict the putative effectors of the species in this study using the default parameters. Furthermore, SCPs were identified using the criteria proposed in [28] (≤400 amino acids and ≥4 cysteine residues).

### 2.9. Plant Materials, Fungal Materials, and Inoculation Assays

*Botryosphaeria dothidea* BDLA16-7 was isolated from Chinese hickory trunk trees in Zhejiang Province, China, showing typical symptoms of canker disease. BDLA16-7 was cultured on potato dextrose agar (PDA) plates at 25 °C in the dark for three days. Then, the mycelial blocks measuring approximately 5 mm × 5 mm were cut for inoculation assays. Shallow incisions, about 2 mm in depth and 5 mm in length, were made on the trunks of two-year-old hickory seedlings using a hole puncher. A single mycelial block was gently pressed against each incision, and then the area was sealed with sterilized tape.

### 2.10. Validation of Gene Expression Using Quantitative qRT-PCR

RNA was isolated using the Qiagen RNeasy Mini Kit (Qiagen Inc., Valencia, CA, USA), followed by cDNA synthesis with the Superscript IV Reverse Transcriptase cDNA Synthesis Kit (TB Green^®^ Premix Ex Taq™ II, Takara Bio Inc., Kusatsu, Japan) using 2 µg of template RNA. All cDNA samples were subsequently diluted to a concentration of 20 ng^−1^ in preparation for qRT-PCR. Gene expression levels were assessed via qRT-PCR employing a Bio-Rad Real-Time PCR System and SYBR Green as the fluorescent dye. The β-tubulin gene from *B. dothidea* (BDLA_00007187) served as the internal reference gene. Primers were generated using Primer3 (https://primer3.ut.ee/, accessed on 5 March 2024), and the specificity of the sequences for the target genes was verified using the NCBI BLASTN web platform (https://blast.ncbi.nlm.nih.gov/Blast.cgi, accessed on 5 March 2024), with the low complexity filter disabled. The aforementioned internal reference gene was utilized to normalize the expression levels of the selected candidate genes. The CDS sequences of β-tubulin gene, candidate genes, and their corresponding primers are provided in Appendix A.

### 2.11. Construction of Deletion Mutants and Complemented Strains

Double-joint PCR was utilized for constructing the deletion vectors of candidate genes. Genomic DNA of wild-type strain BDLA16-7 served as the template for amplifying the sequences flanking the candidate genes. Concurrently, hygromycin resistance (HPH) cassettes were cloned from the pKHT plasmid through amplification with HPH-F/R primers. The amplicons were merged in the second round of PCR, of which the product was used as the template for the final construct’s amplification using nested primers. PEG-mediated protoplast transformation was applied to introduce the vectors into the wild-type BDLA16-7 strain. Selection of potential gene-deletion mutants was conducted on PDA medium augmented with 100 µg/mL of hygromycin. For the complementation assay, segments comprising the gene promoters, green fluorescent protein (GFP), and the coding regions of the candidate genes were amplified and fused by double-joint PCR. The resulting PCR products were transformed with *XhoI*-digested pYF11-RFP plasmid into *Saccharomyces cerevisiae* XK1-25 employing the Alkali-Cation Yeast Transformation Kit (MP Biomedicals, Solon, OH, USA). Recombinant plasmids were extracted from the transformed yeast cells using the Yeast Plasmid Extract Kit (Solarbio, Beijing, China) and subsequently propagated in the *Escherichia coli* strain *DH5*α. These vectors were then reintroduced into the gene-deletion mutants through PEG-mediated transformation.

## 3. Results

### 3.1. Genome Assemblies of Eight Botryosphaeriaceae Species and Features of Note

The genomes of *B. dothidea* strain BDLD16-7 (BDLA), *B. cortices* strain 16-35 (BCTK), *B. fabicerciana* strain 18-2 (BFLG), and *B. qingyuanensis* strain 16-30 (BQTK) were sequenced using an Oxford Nanopore Technologies (ONT) platform with a sequencing depth higher than 100× and were reported in our previous work [29]. Meanwhile, four species of Botryosphaeriaceae with distinct lineages (*M. phaseolina* (MP); *N. parvum* (NP); *D. corticola* (DC); and *L. theobromae* (LT)) were combined for a comprehensive comparison. The similar genome sizes of six Botryosphaeriaceae species, except MP and DC, are provided in Table 1, ranging from 43.69 to 45.98 Mb, and are close to the k-mer analysis results (Appendix A). These results indicate larger genomes for these Botryosphaeriaceae species compared to the average genome size of *Ascomycota* (36.9 Mb) [30].

The assembly quality of BDLA, BQTK, BCTK, and BFLG varied greatly compared to that of MP, NP, DC, and LT due to different sequencing technologies and the assembly toolkit. For example, contig numbers ranging from 13 to 15 were found for BDLA, BCTK, BQTK, and BFLG, with N_50_ values ranging from 3.86 to 3.96 Mb (Table 1). In contrast, MP, NP, DC, and LT were composed of 18 to 296 contigs, with N_50_ values ranging from 0.46 to 4.83 Mb. Obviously higher GC contents of BDLA, BQTK, BCTK, and BFLG (51.90% to 52.81%) were presented relative to MP, NP, DC, and LT (36.40% to 48.42%). The completeness and accuracy of the genome assemblies and gene model predictions within the species were determined via a Benchmarking Universal Single-copy Orthologs (BUSCO) analysis with the ascomycota_odb10 database (1706 genes) as the reference. In a genome-level BUSCO analysis, 1655 to 1674 complete genes of the 1706 single-copy genes were predicted (Figure 1A), which was consistent with the result of a protein-level BUSCO analysis (Figure 1B). This revealed the high completeness of the genome assemblies of the species in this study.

### 3.2. Identification of Repeat Sequences and Protein-Coding Genes

We detected and subsequently masked repetitive elements within the genomes of the eight Botryosphaeriaceae species. Similar repetitive contents were observed in BDLA, BCTK, QTK, and BFLG (6.25% to 8.46%), but quite different to those of MP (17.92%), LT (5.06%), and DC (2.72%) (Table 1, Figure 1C). Tandem repeats, low-complexity sequences, and dispersed transposable elements are the main categories of masked repeats (Appendix A). DNA/TcMar-Fot1 is the dominant subclass family of the class II DNA transposon across BDLA, BCTK, BQTK, BFLG, MP, and NP (Figure 1D). For class I retrotransposons, the subclasses of LTR/Copia, LTR/Gypsy, and LINE/Tad1 are overrepresented across all species (Figure 1E,F). Furthermore, we found that DNA/hAT-Charlie (class II DNA) transposon elements are unique to the BDLA genome.

An ab initio pipeline, coupled with a known gene model dataset retrieved from Swiss-Prot and transcriptome data, was employed for the genome annotations of BDLA, BQTK, BCTK, and BFLG. The protein-coding gene counts per species ranged from 10,366 to 14,845, with similar average length (from 1530 to 1757 bp) and gene density (0.24 to 0.31 genes per kb) values (Table 2). Moreover, lower ratios of exons and introns in BDLA, BQTK, BCTK, and BFLG indicate simpler gene structures compared to MP, NP, DC, and LT.

### 3.3. Synteny Analysis of BDLA Genome Reveals Two Accessory Contigs and Lineage-Specific (LS) Genes

We investigated the extent of the co-linearity of BDLA, BQTK, BCTK, and BFLG, among which the BDLA genome showed high degrees of sequence identity and synteny with BQTK, BCTK, and BGLG (Figure 2A). On the other hand, an intraspecies synteny analysis showed only two pairs of syntenic blocks (contig Bd-9 and contig Bd-3; and contig Bd-10 and contig Bd-4) in the BDLA genome (Figure 2B). This indicates a conservative evolutionary history among these four *Botryosphaeria* species and limited gene duplication events occurring after their divergence from the ancestral lineage. Moreover, two BDLA contigs, Bd-12 and Bd-13, presented no syntenic blocks with closely related *Botryosphaeria* species, which suggested that Bd-12 and Bd-13 might be accessory contigs of the BDLA genome.

Through syntenic comparison, Ref. [31] distinguished lineage-specific (LS) regions in the selected genome, which included genes that play an important role in fungal pathogenicity. Inspired by this, we extracted LS regions from the BDLA genome based on the genome alignment between BDLA, BQTK, BCTK, and BFLG. In Figure 2B, it is evident that the LS regions in BDLA are mainly distributed on the accessory contigs of Bd-12 and Bd-13. Notably, these LS regions tend to cluster towards the contig terminals, exhibiting a high repeat density but a lower GC content. In BDLA, a total of 501 genes exist in LS regions (LS genes), with the highest count in Bd-13 (83 genes), followed by Bd-14 (59 genes) and Bd-04 (57 genes) (Figure 2C).

A Gene Ontology (GO) enrichment analysis of the LS genes showed that the enriched GO terms mainly include the biosynthesis process of nucleic acid and the metabolic processes of nitrogen compounds, organic compounds, and cellular macromolecules (Figure 2D). In total, 164 of 501 LS genes carry an annotated domain (Appendix A). Among them, 22 LS genes encode viral integrase (rve: PF00665), 9 encode zinc finger transcriptional factors (zf-CCHC: PF00098), 8 encode orsellinic acid (OrsD: PF12013), 7 encode kinase (Pkinase: PF00069), 6 encode cytochrome P450 (p450: PF00067), and 2 encode major facilitator superfamily proteins (MFS_1: PF07690) (Figure 2E).

### 3.4. Ortholog Construction of Botryosphaeriaceae Species Uncovered Rapid Expanded Ortholog in BDLA

We established orthologous relationships between BDLA, BCTK, BFLG, BQTK, MP, NP, DC, and LT using OrthoFinder v2.5.5, and 96,176 (96.11%) of the 100,070 total genes were assigned to 15,050 orthologs (Appendix A), which ranged in size from 2 to 38 genes. Overall, 2122 genes (2.12%) were assigned to 657 species-specific orthologs (Figure 3A, Appendix A). Among them, 868 genes in MP were assigned to 261 species-specific orthologs, with LT, NP, and DC following closely (Appendix A). In contrast, only 21 to 37 genes were assigned to species-specific orthologs in BDLA, BQTK, BCTK, and BFLG. In total, 6785 core orthologs, comprising genes from each species in this study, were detected, and 6064 of them were identified as single-copy orthologs (Figure 3A, Appendix A). In addition to species-specific and core orthologs, we also investigated multiple-species-involved (MSI) orthologs, which included members of two to seven species. For instance, 378 two-species MSI orthologs of BDLA suggest that genes from another species, apart from BDLA, were arranged in these orthologs (Figure 3B). As a result, the ratios of the four-species (891), five-species (975), six-species (1141), and seven-species (1758) MSI orthologs were higher than those of the two-species (378) and three-species (375) MSI orthologs in BDLA. Similar conditions were also found in other Botryosphaeriaceae species, which indicates conservative orthologous relationships between these species and supports the synteny analysis results.

We established the phylogenic relationships of the Botryosphaeriaceae species in this study using the detected single-copy orthologs. The phylogenetic tree was divided into two clades: one clade included BDLA, BCTK, BQTK, and BFLG, indicating close genetic relatedness, but displayed evolutionary distance from MP, NP, DC, and LT. In detail, BDLA exhibited a close phylogenetic relationship with BQTK, while BCTK was grouped alongside BFLG.

We further detected rapidly evolving orthologs using CAFÉ v5.0, which helped explore the significant changes in these Botryosphaeriaceae species compared to their most recent common ancestor (MRCA). The calculated lambda values were automatically set to 0.0001, and orthologs with Viterbi *p*-values lower than 0.01 were defined as rapidly evolving. In Figure 3C, 579 orthologs (+29 orthologs expanded/−550 orthologs contracted) were detected in node <7>, the MRCA of BDLA, BCTK, BQTK, and BFLG; 377 orthologs (+127/−250) were detected in BDLA; 676 orthologs (+55/−621) were detected in BQTK; 300 orthologs (+91/−209) were detected in BCTK; and 330 orthologs (+55/−275) were detected in BFLG. However, more rapidly evolving orthologs were identified in MP (1772: +551/−1221), NP (1772: +220/−2663), DC (1772: +139/−1614), and LT (1200: +573/−627). This indicates less divergency in the genomes of BDLA, BCTK, BQTK, and BFLG.

To explore *B. dothidea*’s pathogenicity in Chinese hickory, genes arranged in rapidly expanded orthologs in BDLA were subjected to a function enrichment analysis, and a total of 296 BDLA genes were detected (Appendix A). Twenty of these genes carry the domain of Chromo (CHRromatin Organization MOdifier) (PF00385); eight genes are involved in orsellinic acid biosynthesis (OrsD: PF12013); seven encode heterokaryon incompatibility proteins (HET: PF06985); six encode cytochrome P450 (p450: PF00067); and five encode major facilitator superfamily proteins (MFS_1: PF07690) (Figure 3D). A GO enrichment analysis of these genes revealed that the enriched GO terms were mainly divided into three categories (Figure 3E): (i) DNA metabolic processes (such as ‘DNA biosynthetic process’ and ‘DNA polymerase activity’); (ii) nitrogen compound metabolic processes; and (iii) aromatic or organic cyclic compound biosynthesis processes.

### 3.5. Positively Selected Genes (PSGs) of BDLA may Be Involved in Response to Xenobiotic Stimulus

Positive selection genes (PSGs), with a non-synonymous/synonymous substitutions ratio (ω = dN/dS) higher than 1, may contribute to the emergence of novel biological functions [32]. To explore the PSGs of *B. dothidea*’s pathogenicity, we calculated the dN/dS ratio for 6064 single-copy orthologs. The calculations were performed using the branch model of the PAML (phylogenetic analysis maximum likelihood)-CODEML algorithm. In this model, the genes of BDLA, BQTK, BCTK, and BFLG within each single-copy ortholog were identified as the foreground branch, respectively. The PSGs were characterized by a dN/dS ratio exceeding 1 and a chi2 value below 0.05. As a result, 32, 48, 25, and 27 PSGs were identified in BDLA, BQTK, BCTK, and BFLG, constituting 0.53%, 0.79%, 0.41%, and 0.45% of the total single-copy genes (Figure 4A, Appendix A).

We compared orthologs containing the PSGs of BDLA, BQTK, BCTK, and little intersection was observed (Figure 4B), which suggests their obvious sequence divergency. Therefore, we subsequently focused on BDLA-specific PSGs. These PSGs were predominantly enriched in GO terms such as ‘response to chemical’, ‘cellular response to chemical stimulus’, and ‘response to xenobiotic stimulus’, with an average dN/dS ratio ranging from 592.06 to 726.35 (Figure 4C). Notably, one of the BDLA PSGs, BDLA_00004043-RA assigned to OG0003604, carries the LRR_8 (PF13855) domain (Figure 4D, Appendix A), which is consistent with research on the NBS-LRR genes of *Arachis duranensis* and *Arachis ipaënsi* [33]. In addition, BDLA_00009721-RA and BDLA_00011736-RA, assigned to OG0005150 and OG0007268, respectively, underwent positive selection pressure (dN/dS ratio of 1.76 and 70.2467) and carry Sec_1 and Sec7-N domains, which were reported to be involved in SNARE complex assembly and vesicle trafficking regulation [34,35]. Moreover, coding genes of one chitinase (OG0005721: BDLA_00008528-RA-containing domain of Glyco_hydro_18) and one chitin biosynthesis enzyme (OG0004492: BDLA_00004853-RA-containing domain of Chitin_synth_2) also had a dN/dS ratio of 999, which implies that positive selection may affect the chitin metabolic process of *B. dothidea*.

### 3.6. Predicted Secretome Comparison Gives Insight into BDLA-Specific Small Secret Cysteine-Rich Proteins (Effectors)

Plant pathogens deploy an arsenal of secreted proteins to interfere with the plant immune system, enabling the successful colonization of the host and infection [36]. BDLA, BCTK, BFLG, and BQTK exhibited similar numbers of secreted proteins, ranging from 997 to 1049, and 807 to 1153 secreted proteins were found in MP, NP, DC, and LT (Appendix A, Figure 5A). The lengths of the identified secreted proteins in all species were primarily distributed in the ranges of 100 to 400 amino acids (aa) and 400 to 1000 aa (Figure 5A). Among these secreted proteins, there were fewer putative effectors in BDLA (83), BCTK (83), BQTK (75), and BFLG (87) compared to the four other Botryosphaeriaceae species, ranging from 78 to 130 (Appendix A). A large proportion of the putative effectors were apoplastic (Figure 5B), and almost all of them were small cysteine-rich proteins (SCPs) (Figure 5C, Appendix A), which suggests that these cysteine-rich effectors may confer *B. dothidea*’s ability to overcome the immune system of its host [37]. The GO enrichment analysis of *B. dothidea*’s putative cysteine-rich effectors provided the insight that these putative effectors are mainly enriched in carbohydrate metabolism (such as pectin, galacturonan, and polysaccharide) and the xenobiotic stimulus response (Figure 5D).

Homology searches were conducted for the secretomes of *B. dothidea* and other Botryosphaeriaceae species. As shown in Figure 5E, the BDLA secretome exhibited a significantly higher identity with those of BCTK, BFLG, and BQTK compared to MP, NP, DC, and LT. Interestingly, we noticed that there were two groups of BDLA-specific secreted proteins, of which 22 were cysteine-rich and 6 were putative effectors. Through a BLAST search against the Pathogen Host Interactions (PHI) database (http://www.phi-base.org/, accessed on 10 November 2023), it was found that eight BDLA-specific SCPs may affect the pathogen’s virulence and five of them carry domains of Glyco_hydro_18, Glyco_hydro_61, Glyco_hydro_16, Glyco_hydro_10, and Glycos_transf_1 (Table 3), which may play a role in carbohydrate metabolism and host cell wall degradation.

### 3.7. Identification of Horizontal Gene Transfer (HGT) Events in BDLA Genome

Horizontal gene transfer is a process where genomic DNA segments are exchanged between organisms that are not in a parent–offspring relationship, which may introduce novel functions and improve the adaptive ability of the recipient organism [38]. We performed a genome-wide HGT event prediction for BDLA to explore its pathogenicity influenced by HGTs. The Alien Index (AI), comparing the best hits from closely related taxa (ingroup) and distantly related taxa (donor) based on their BLAST E-values [39], was calculated for each protein-coding gene of BDLA using the AvP toolkit [27], coupled with support from phylogenetic evidence.

As shown in Table 4, 42 HGT events were identified in BDLA. The most likely donor species are bacteria for HGT_1 to HGT_7, fungi for HGT_8 to HGT_31, other eukaryotes for HGT_32 to HGT_35, oomycota for HGT_36 to HGT_39, and viridiplantae for HGT_40 to HGT_42. This indicates that genetic exchange between members of the *B. dothidea* fungus community is particularly frequent. Among the fungi-derived genes, BDLA_00005439-RA (HGT_16) is a putative cellulase (including a Glyco_hydro_7 domain) that may be involved in host cell wall degradation. BDLA_00001191-RA (HGT_12) has GTPase activity with the Ras domain, which may be involved in external signal transduction. Moreover, one GDSL-like Lipase (BDLA_00002083-RA, HGT_22), one Peptidase (BDLA_00002724-RA, HGT_23), one ctr copper transporter (BDLA_00007786-RA, HGT_10), one Jacalin-like lectin (BDLA_00010741-RA, HGT_19), and one Molybdate transporter protein (BDLA_00007280-RA, HGT_21) were also observed. The bacteria-derived HGTs included one thymidine phosphorylase (BDLA_00012548-RA, HGT_5) and one Alkyl sulfatase (BDLA_00003435-RA, HGT_6). Moreover, one Pyridoxal-phosphate-dependent enzyme (BDLA_00012074-RA, HGT_36) was transferred from the oomycete kingdom.

The PHI base annotations revealed that BDLA_00008949-RA (HGT_3), BDLA_00010862-RA (HGT_11), BDLA_00001191-RA (HGT_12), and BDLA_00010741-RA (HGT_19) might affect pathogenicity. Notably, BDLA_00011030-RA (HGT_17) is a putative avirulence effector and carries a domain of serine aminopeptidase, which suggests that this gene might be involved in the inhibition of the host immune system through its protease activity.

### 3.8. Secreted Glucanase, Orsellinic Acid Biosynthesis Enzyme, and MFS Transporters Play an Important Role in Pathogenicity of B. dothidea

The bioinformatics analysis above identified several gene families that might be related to *B. dothidea*’s pathogenicity. Here, we performed qRT-PCR assays to validate the expression level of genes assigned to these families. We found that one secreted glucanase (BDLA_00012709), two secreted chitinase (BDLA_00008528 and BDLA_00004853), two orsellinic acid biosynthesis enzymes (BDLA_00007003 and BDLA_00010036), three major facilitator superfamily proteins (BDLA_00004985, BDLA_00007512 and BDLA_00010781), one viral integrase (BDLA_00005840), and one LRR-containing protein (BDLA_00004043) significantly upregulated during the infection stage. As Figure 6 shows, BDLA_00012709, BDLA_00010036, BDLA_00007003, BDLA_00008528, and BDLA_00010781 were specifically upregulated at 8 days post-inoculation (dpi). Meanwhile, BDLA_00004985, BDLA_00007512, BDLA_00004853, BDLA_00004043, and BDLA_00015840 exhibited a consistent upregulation trend in expression. Among them, one glucanase (BDLA_00012709), one orsellinic acid biosynthesis enzyme (BDLA_00007003), and two MFS transporters (BDLA_00010781 and BDLA_00004985) showed the highest upregulated expression level and were chosen to evaluate their pathogenic ability on hickory trunk. Disease lesions caused by knockout strains of these four pathogenic candidates were obviously smaller than those caused by wild-type and their corresponding complemented strains (Figure 7). These results suggest that glucanase, orsellinic acid biosynthesis enzymes, and MFS transporters play an important role in *B. dothidea*’s pathogenicity. According to the bioinformatics analysis above, BDLA_00007003 and BDLA_00010781 were located in the lineage-specific (LS) regions of *B. dothidea* (Appendix A). BDLA_00004985 is assigned to ortholog group OG0000329, which is specifically expanded in *B. dothidea* (Appendix A). Moreover, BDLA_00012709 is one small secret cysteine-rich protein specific in *B. dothidea* (Table 3) and is a putative virulence protein according to the annotation based on the PHI database (Appendix A).

## 4. Discussion

Trunk cankers pose a significant threat to Chinese hickory production in the Lin’an district of Zhejiang Province, China [21]. In our previous research, the majority of the pathogenic isolates belonged to *B. dothidea* wild-type strain BDLA16-7. This highlighted *B. dothidea*’s enhanced pathogenicity and host colonization abilities, particularly in Chinese hickory. This motivated us to explore the potential mechanisms underlying this phenomenon through comparative genomics between *B. dothidea* BDLA16-7 (BDLA) and other closely related species.

Higher GC contents were observed in the genomes of BDLA, BCTK, BQTK, and BFLG compared to MP, NP, DC, and LT. According to [40], increased GC contents in genomic regions promote transcriptional activity. We speculated that in BDLA, BCTK, BQTK, and BFLG, higher GC contents might be associated with genes related to their infection of Chinese hickory.

Expectedly high degrees of sequence identity and synteny between BDLA, BQTK, BCTK, and BFLG were described, which was consistent with [22]. This suggests a conserved genomic structure and content among *Botryosphaeria* species, indicating gradual evolution. This result contrasts with that of destructive filamentous fungi, such as *Phytophthora infestans* and *Magnaporthe oryzae*, which can rapidly kill host plants, prompting a rapid evolutionary ‘arms race’ and shaping divergent genomes among closely related species [41,42]. However, we still found two specific contigs and other lineage-specific (LS) regions in the BDLA genome, which include genes that may be related to *B. dothidea*’s stronger pathogenicity in Chinese canker. A significant proportion of BDLA genes in LS regions were occupied by genes encoding virus integrase, which is responsible for the viral replication that catalyzes the covalent integration of viral cDNA into the host genome [43]. It is well known that mycoviruses can attenuate the growth and virulence of *B. dothidea* [44]. However, our HGT analysis revealed that these virus integrases were not acquired from viruses or other distant species. One possible explanation is that these virus integrases originated from *B. dothidea*’s ancestors and might indicate to *B. dothidea*’s pathogenicity by shaping the endophytic lifestyle of *B. dothidea*. Such integrases may be found in retrotransposons [45,46]. Interestingly, we found that heterokaryon incompatibility (HI) proteins were expanded in BDLA as well. Heterokaryon incompatibility (HI) is a non-self-recognition phenomenon in filamentous fungi, and it plays an important role in limiting resource plundering and restricting viral transfer between strains [47,48]. Therefore, *B. dothidea* also possesses sufficient capability to restrict viral transfer. Taken together, we speculate that there is a ‘viral-balance’ system in *B. dothidea*’s genome, which suggests that *B. dothidea* may control the orderly integration of viruses or retrotransposons into its genome for environmental adaptation. However, detailed mechanisms still need further research in the future. Additionally, we also found that two MFS transporter coding genes exists in LS regions and deleting one of them (BDLA_00010781) reduced *B. dothidea*’s pathogenicity. Combined with previous studies [49,50], we proposed two possibilities: (i) BDLA_00010781 is involved in fungal toxin secretion; (ii) BDLA_00010781 is important for hyphal morphology and conidiation, which affect fungal pathogenicity.

An overrepresentation of genes was involved in orsellinic acid biosynthesis and carried the orsD domain (PF12013) in the BDLA-specific gene set. Endophytic fungi can generate a range of bioactive metabolites to protect the hosts against other pathogens and herbivores in harsh environments for the purpose of nutrient absorption from host plants [51,52,53]. For instance, interactions with bacteria induce *Aspergillus nidulans* to produce the archetypal polyketide orsellinic acid [54]. In [55], the authors utilized orsellinic acid, a compound isolated from the endophytic fungus *Epicoccum Nigrum*, to produce biocompatible green silver nanoparticles. These nanoparticles exhibited significant antifungal activity against *Alternaria solani*. In this study, we found BDLA_00007003, containing the orsD domain, significantly upregulated since 3 dpi, and positively regulated *B. dothidea*’s pathogenicity. It will be interesting to infer that this orsellinic acid biosynthesis coding gene might be induced to inhibit its fungal competitors in diseased Chinese hickory samples.

An analysis of positively selected genes (PSGs) in BDLA revealed that these PSGs are mainly involved in the response to xenobiotic stimuli. Specifically, BDLA_00003994-RA, assigned to OG0003633, encodes one LRR-containing protein. In plants, this type of protein is well known for regulating the immune system [56]. However, LRR-containing proteins are also involved in development and pathogenicity in *Phytophthora sojae* [57,58]. Thus, the LRR-containing protein of BDLA may undergo selection pressure because it affects *B. dothidea*’s development and pathogenicity. Meanwhile, one Sec_1-containing protein and one Sec_2-containing protein of BDLA were also putative PSGs, and they may affect protein trafficking.

The putative secretome of *B. dothidea* shows higher similarity with BCTK, BQTK, and BFLG compared to MP, NP, DC, and LT. Small secreted cysteine-rich proteins (SCPs) have been implicated as key virulence factors in fungal pathogens, contributing to the establishment of colonization in host plants [59]. In the current study, we observed that some of the BDLA-specific SCPs encode chitinase and have undergone positive selection. This association may be related to the degradation of the cell wall in fungal competitors of *B. dothidea*, considering the absence of chitin content in the host plants. We speculate that this result may partially explain the dominant role of *B. dothidea* in diseased trunk samples. Furthermore, we also noticed that one secreted glucanase, BDLA_00012709, might enhance fungal pathogenicity by degrading the cell wall of the host.

In this study, we presented a high-quality genome assembly of *B. dothidea* and conducted an in-depth analysis of the relationship between *B. dothidea*’s genome and its pathogenicity in Chinese hickory. Using reported genomic data of *B. dothidea* [22], we identified and verified one glucanase and several chitinase that are pathogenicity-associated. On the other hand, we noticed that orsellinic acid biosynthesis-associated enzymes of *B. dothidea* may affect its competitiveness in the fungal community of the diseased samples, which was not reported before. Two MFS transporters were also validated for their function during the procession of *B. dothidea*’s infection. Moreover, we speculate that the viral integrase and heterokaryon incompatibility proteins of *B. dothidea* may collaborate to regulate the integration of viruses into the *B. dothidea* genome. This coordination could potentially influence *B. dothidea*’s pathogenicity and affect shaping its endophytic lifestyle.

This study shed light on the evolutionary process of *B. dothidea* and other closely related species isolated from diseased Chinese hickory trees, and several candidates might be related to *B. dothidea*’s pathogenicity and colonization ability. However, the limitations of this comparative analysis are (1) it will be difficult to predict genes located in complex or high-repeat regions without the support of chromosome-level genome assemblies; (2) functional annotations for each gene are based only on sequence similarity to known domains, but they cannot identify key sites without in-depth experimental validation, which sometimes is more critical to gene functions. Hence, in-depth elucidation of the mechanisms is essential for a comprehensive understanding of *B. dothidea*’s pathogenicity in the future.

## Figures and Tables

**Figure 1 jof-10-00299-f001:**
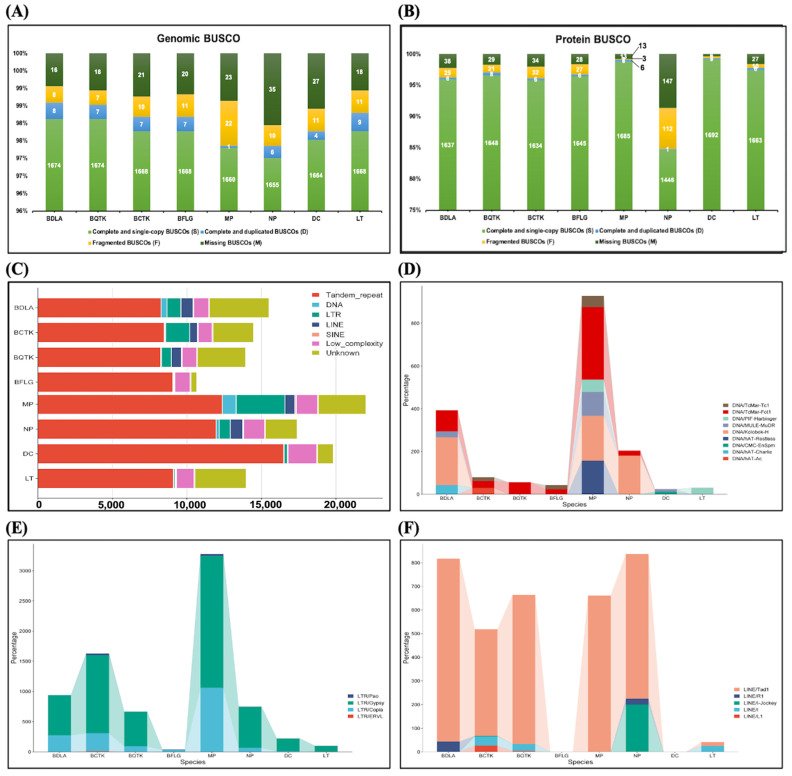
Genome assembly assessment and repeat sequence detection of Botryosphaeriaceae species. Genome assembly assessment performed by BUSCO at genome level (**A**) and protein level (**B**). (**C**–**F**) provide distributions of repeat sequences across Botryosphaeriaceae species: (**C**) total repeat sequences; (**D**) class II DNA transposons; (**E**) class I retrotransposons of LTR subclass; and (**F**) class I retrotransposons of LINE subclass.

**Figure 2 jof-10-00299-f002:**
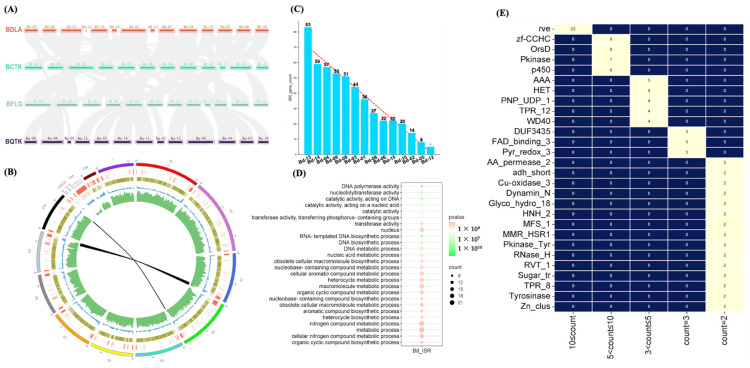
Synteny analysis of BDLA and investigation of genes within BDLA-specific regions. (**A**) Inter-species synteny analysis between BDLA, BQTK, BCTK, and BFLG; (**B**) intra-species synteny analysis and genome content annotations of BDLA: red bar lineage-specific (LS) regions, earthy yellow heatmap means GC content, blue peak diagram represents sequence density, and green means gene density. (**C**) Distribution of genes within LS regions on different contigs in BDLA genome. (**D**) GO enrichment analysis of BDLA genes within LS regions; (**E**) Pfam domain annotation of BDLA genes within LS regions.

**Figure 3 jof-10-00299-f003:**
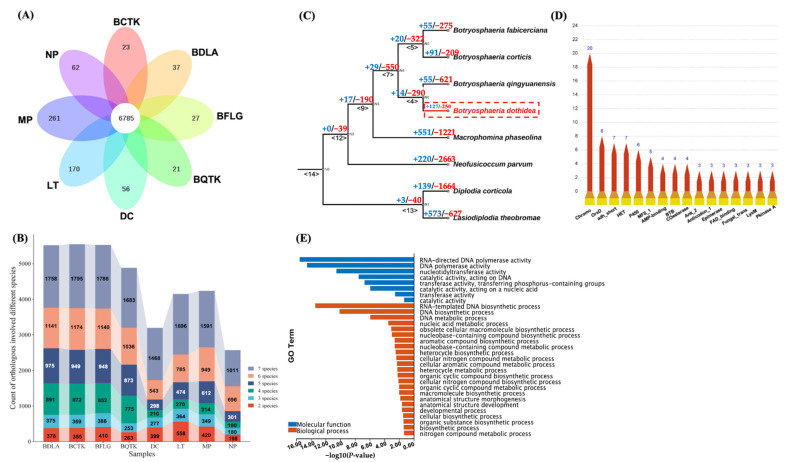
Ortholog analysis and identification of rapidly evolving orthologs. (**A**) Orthologous relationships between BDLA, BCTK, BFLG, BQTK, MP, NP, DC, and LT. (**B**) Analysis of species associated with orthologs, which was used for their conservation. (**C**) Phylogenic analysis and rapidly evolving orthologs of BDLA, BCTK, BFLG, BQTK, MP, NP, DC, and LT. Blue numbers represent counts of expanded orthologs, and red numbers represent contracted orthologs. (**D**) GO enrichment analysis of genes assigned to expanded orthologs in BDLA. (**E**) Pfam domain annotation of genes assigned to expanded orthologs in BDLA.

**Figure 4 jof-10-00299-f004:**
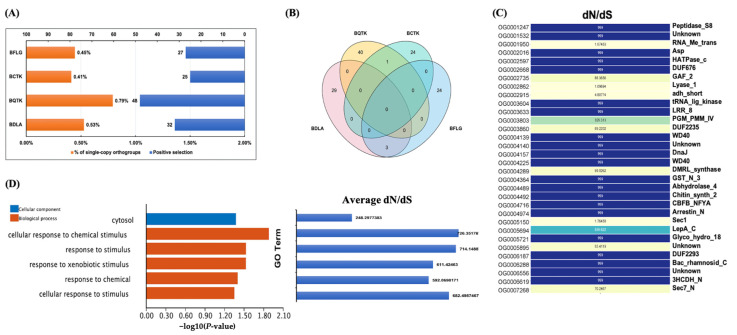
Investigation of positively selected genes in BDLA, BQTK, BCTK, and BFLG. (**A**) Proportion of positively selected genes among all single-copy orthologs. Blue bars represent numbers of positively selected genes, and orange bars indicate their corresponding proportions among all single-copy orthologs. (**B**) Venn diagram showing overlaps of positively selected genes in BDLA, BQTK, BCTK, and BFLG. (**C**) Pfam annotations and dN/dS ratios of positively selected genes of BDLA and their corresponding ortholog IDs. (**D**) GO enrichment analysis of positively selected genes of BDLA and their corresponding average dN/dS ratios.

**Figure 5 jof-10-00299-f005:**
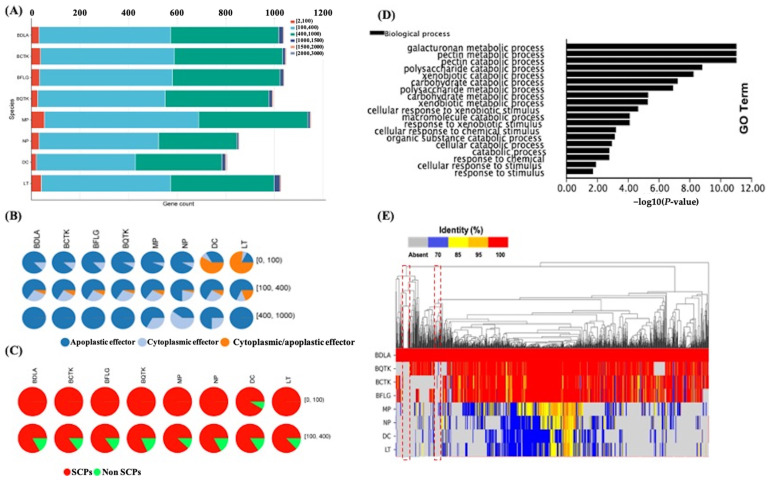
Secretome analysis of Botryosphaeriaceae species. (**A**) Counts of secreted proteins in each Botryosphaeriaceae species and their corresponding length distribution. Different colors represent sequence length intervals. (**B**) Distribution of lengths and categories of putative effectors in each Botryosphaeriaceae species. The numbers on the right represent sequence length intervals. (**C**) Proportion of small secreted cysteine-rich proteins (SCPs) and their corresponding length distribution. The numbers on the right represent sequence length intervals. (**D**) GO enrichment analysis of *B. dothidea*’s putative cysteine-rich effectors. (**E**) Homologous search. Red dashed box marked secret proteins specific in BDLA.

**Figure 6 jof-10-00299-f006:**
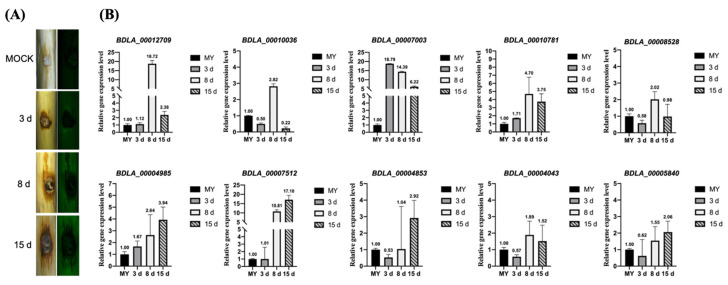
Assessment of *B. dothidea* BDLA16-7’s pathogenicity. (**A**) Disease symptoms on Chinese hickory trunk caused by *B. dothidea* BDLA16-7 at 3, 8, and 15 days post-inoculation (dpi); (**B**) quantitative real-time PCR verification of expression of selected coding genes during infection stage.

**Figure 7 jof-10-00299-f007:**
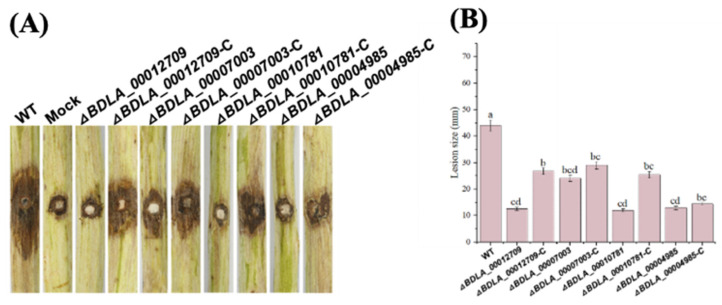
BDLA_00012709, BDLA_00007003, BDLA_00010781, and BDLA_00004985 are required for BDLA16-7’s pathogenicity. (**A**) Disease symptoms on Chinese hickory trunk caused by each strain for 15 days post-inoculation (dpi); (**B**) lesion size on Chinese hickory trunk caused by each strain. a, b, c, and d represent the significant difference (one-way ANOVA test, *p* < 0.01).

**Table 1 jof-10-00299-t001:** Genome features of genome assemblies of Botryosphaeriaceae species.

Species	Contigs	N50 (Mb)	Assembly Length	Repeat Percent (%)	Predicted Genes	GC Content (%)
BDLA	15	3.86	45.98	8.2	13,052	52.27
BQTK	14	3.93	44.31	6.25	12,141	52.81
BCTK	13	3.96	45.13	8.46	12,910	51.9
BFLG	13	3.92	44.87	6.51	12,863	51.99
MP	75	4.83	50.55	17.92	14,845	36.4
NP	18	2.54	43.99	6.89	10,366	48.42
DC	181	0.46	34.99	5.06	10,839	43.23
LT	296	0.88	43.69	2.72	13,054	42.08

**Table 2 jof-10-00299-t002:** Statistic of protein-coding genes in each Botryosphaeriaceae species.

Annotation Statistic	BDLA	BQTK	BCTK	BFLG	MP	NP	DC	LT
Average gene length (bp)	1660	1757	1601	1649	1750	1530	1747	1677
Total length of CDS (Mb)	18.51	17.71	18.1	18.26	20.15	13.21	17.23	19.5
% of genome covered by genes	47.13	48.14	45.79	47.27	51.38	36.06	54.13	50.11
Gene density (gene/kb)	0.28	0.27	0.29	0.29	0.29	0.24	0.31	0.3
Exons per gene	2.53	2.41	2.51	2.55	2.97	3.16	2.99	3.08
Introns per gene	1.53	1.42	1.51	1.55	1.96	2.16	1.99	2.08

**Table 3 jof-10-00299-t003:** Annotations of BDLA-specific small secret cysteine-rich proteins (SCPs).

Gene ID	Protein Length	Cysteine Proportion	SCP Identification	EffectorP Prediction	PHI Annotation	Pfam Domain	Function Annotation
BDLA_00001985-RA	111	6.25	Y	Apoplastic effector	-	-	-
BDLA_00002075-RA	227	0.88	Y	-	effector_(plant_avirulence_determinant)	Glyco_hydro_61	Fungal cellulose-binding domain-containing protein
BDLA_00002749-RA	306	1.63	Y	-	-	-	-
BDLA_00003112-RA	189	1.05	Y	-	-	-	-
BDLA_00003469-RA	290	0.34	Y	-	reduced_virulence	TB2_DP1_HVA22	Belongs to the type-B carboxylesterase lipase family
BDLA_00003643-RA	393	0.76	Y	-	loss_of_pathogenicity	p450	Belongs to the cytochrome P450 family
BDLA_00006476-RA	292	3.07	Y	-	-	-	-
BDLA_00007123-RA	140	4.26	Y	-	-	-	-
BDLA_00007170-RA	234	6.38	Y	Apoplastic effector	-	-	-
BDLA_00007782-RA	102	0.97	Y	Cytoplasmic effector	-	-	-
BDLA_00008471-RA	318	0.94	Y	-	-	-	-
BDLA_00008838-RA	128	3.88	Y	-	-	-	-
BDLA_00009176-RA	284	3.51	Y	-	-	-	-
BDLA_00009481-RA	136	1.46	Y	-	-	SnoaL_2	Snoal-like polyketide cyclase family protein
BDLA_00009719-RA	345	1.45	Y	-	reduced_virulence	Glyco_hydro_10	glycoside hydrolase family 10 protein
BDLA_00009878-RA	124	1.6	Y	-	reduced_virulence	Glycos_transf_1	Starch synthase catalytic domain(ags1)
BDLA_00010018-RA	90	8.79	Y	Apoplastic effector	-	-	-
BDLA_00011085-RA	73	8.11	Y	Apoplastic effector	-	-	-
BDLA_00011913-RA	209	0.48	Y	-	reduced_virulence	Lipase_GDSL_2	GDSL-like Lipase/Acylhydrolase family
BDLA_00012014-RA	152	8.5	Y	-	-	-	-
BDLA_00012024-RA	287	0.69	Y	-	-	Peptidase_S15	X-Pro dipeptidyl-peptidase
BDLA_00012709-RA	383	2.08	Y	-	reduced_virulence	Glyco_hydro_16	glycoside hydrolase family 16 protein
BDLA_00012987-RA	233	2.56	Y	Apoplastic effector	-	-	-
BDLA_00012997-RA	304	2.62	Y	-	-	-	-
BDLA_00013002-RA	270	1.48	Y	-	-	DUF3455	Protein of unknown function (DUF3455)
BDLA_00013015-RA	274	1.09	Y	-	increased_virulence_(hypervirulence)_reduced_virulence	Glyco_hydro_18	Belongs to the glycosyl hydrolase 18 family(CHT2)

**Table 4 jof-10-00299-t004:** Annotations of HGT genes identified in BDLA.

Gene ID	HGT Event ID	Pfam Domain	Putative Function	Best Hit on PHI-Base (E-Value)	Most Likely Donor
BDLA_00000792-RA	HGT_1	Cupin_3	Protein of unknown function (DUF861)	-	Bacteria
BDLA_00005852-RA	HGT_2	-	-	Q9HV44; heat shock protein; reduced virulence	Bacteria
BDLA_00004793-RA	HGT_3	-	-	A0A098DK85; protease; unaffected pathogenicity	Eukaryota
BDLA_00003727-RA	HGT_4	HRI1	Protein HRI1	-	Fungi
BDLA_00007369-RA	HGT_5	Methyltransf_11	ubiE/COQ5 methyltransferase family	-	Fungi
BDLA_00007786-RA	HGT_6	Ctr	ctr copper transporter family protein(ctr4)	B0XUP5; copper transporters; unaffected pathogenicity	Fungi
BDLA_00010862-RA	HGT_7	Peptidase_M43	Pregnancy-associated plasma protein-A	C5P3X6; Metalloproteinase; reduced virulence	Fungi
BDLA_00001191-RA	HGT_8	Ras	GTPase activity	A0A139Y2L7; small GTPase; reduced virulence	Fungi
BDLA_00003074-RA	HGT_9	Acetyltransf_7	Acetyltransferase (GNAT) domain	-	Fungi
BDLA_00010710-RA	HGT_10	EHN	Epoxide hydrolase N terminus	-	Fungi
BDLA_00010289-RA	HGT_11	Zn_clus	Oxidoreductase NAD-binding domain	Q4W9X3; flavohemoglobins; unaffected pathogenicity	Fungi
BDLA_00005439-RA	HGT_12	Glyco_hydro_7	Belongs to the glycosyl hydrolase 7 (cellulase C) family	G4ZRT3; conserved glycoside hydrolase family 7 cellobiohydrolase; reduced virulence	Fungi
BDLA_00011030-RA	HGT_13	Abhydrolase_1	Serine aminopeptidase, S33	Q8PC98; effector protein; effector (plant avirulence determinant)	Fungi
BDLA_00007899-RA	HGT_14	DAO	FAD dependent oxidoreductase	-	Fungi
BDLA_00010741-RA	HGT_15	Jacalin	Jacalin-like lectin domain	P9WKQ1; extracellular nuclease; reduced virulence	Fungi
BDLA_00010678-RA	HGT_16	DUF1349	Protein of unknown function (DUF1349)	-	Fungi
BDLA_00007280-RA	HGT_17	MFS_MOT1	Molybdate transporter of MFS superfamily	-	Fungi
BDLA_00002083-RA	HGT_18	Lipase_GDSL_2	GDSL-like Lipase/Acylhydrolase family	-	Fungi
BDLA_00002724-RA	HGT_19	Peptidase_S58	Peptidase family S58	-	Fungi
BDLA_00007942-RA	HGT_20	-	-	-	HGT_complex
BDLA_00003219-RA	HGT_21	RVT_2	Mitochondrial protein	-	NM_Eukaryota_complex
BDLA_00005227-RA	HGT_22	-	-	-	NM_Eukaryota_complex
BDLA_00012074-RA	HGT_23	PALP	Pyridoxal-phosphate-dependent enzyme	G2XJR0;1-Aminocyclopropane-1-carboxylate deaminase; increased virulence (hypervirulence)	Oomycota
BDLA_00008930-RA	HGT_24	NAD_binding_8	FAD-binding domain	Q5GFD3; Mannitol 1-phosphate dehydrogenase; unaffected pathogenicity	Oomycota
BDLA_00008949-RA	HGT_25	Ank_4	spectrin binding	J9VLD1; Cyclin-dependent protein kinase inhibitor; reduced virulence	Bacteria
BDLA_00004977-RA	HGT_26	-	Podospora anserina S mat genomic DNA chromosome	-	Bacteria
BDLA_00012548-RA	HGT_27	PYNP_C	thymidine phosphorylase activity (TYMP)	-	Bacteria
BDLA_00003435-RA	HGT_28	Lactamase_B	Alkyl sulfatase dimerization (MA20_17395)	U3M7S2; heat-sensitive alkyl sulphatase; unaffected pathogenicity	Bacteria
BDLA_00003351-RA	HGT_29	-	-	Q8J286; copper transporting ATPase; loss of pathogenicity	Bacteria
BDLA_00006818-RA	HGT_30	-	-	J9VLD1; cyclin-dependent protein kinase inhibitor; reduced virulence	Eukaryota
BDLA_00008015-RA	HGT_31	UDPGT	Belongs to the OSBP family	-	Fungi
BDLA_00007233-RA	HGT_32	-	-	-	Fungi
BDLA_00010519-RA	HGT_33	-	-	-	Fungi
BDLA_00002024-RA	HGT_34	-	-	-	Fungi
BDLA_00008349-RA	HGT_35	-	-	-	Fungi
BDLA_00001002-RA	HGT_36	-	-	-	Fungi
BDLA_00009177-RA	HGT_37	-	-	-	Fungi
BDLA_00010857-RA	HGT_38	-	-	-	Fungi
BDLA_00004645-RA	HGT_39	-	-	-	Oomycota
BDLA_00012380-RA	HGT_40	-	-	-	Oomycota
BDLA_00012910-RA	HGT_41	-	-	-	Viridiplantae
BDLA_00002811-RA	HGT_42	-	-	-	Viridiplantae
BDLA_00001009-RA	HGT_43	-	-	-	Viridiplantae

## Data Availability

All data in this paper have been deposited in the Genome Warehouse (GWH, https://ngdc.cncb.ac.cn/gwh, accessed on 4 April 2023) in the National Genomics Data Center, China National Center for Bioinformation (CNCB-NGDC Members and Partners, 2021), under accession number GWHBEBO00000000. The raw sequence reads are also publicly accessible at the Genome Sequence Archive (GSA, https://ngdc.cncb.ac.cn/gsa/, accessed on 4 April 2023) in CNCB-NGDC under accession number CRA004612 (BioProject: PRJCA005744).

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
