# Peer review of "The Comparative Genomics of Botryosphaeriaceae Suggests Gene Families of Botryosphaeria dothidea Related to Pathogenicity on Chinese Hickory Tree"

_jof, 2024, doi:10.3390/jof10040299_

Round 1

Reviewer 1 Report

The research manuscript discusses the comparative genomics of Botryosphaeriaceae, focusing on the gene families in Botryosphaeria dothidea related to its pathogenicity on Chinese hickory trees. The study presents a detailed analysis of high-quality genome assemblies, revealing syntenic relationships between B. dothidea and related species and the conservative evolution within the Botryosphaeriaceae family. It identifies specific genes in B. dothidea that are either unique or expanded, suggesting their role in the pathogen's ability to infect Chinese hickory trees. These include genes encoding for viral integrases, heterokaryon incompatibility proteins, secreted chitinases, and enzymes for orsellinic acid biosynthesis. The aims of this study is clear and the results in interest to me. I don’t have major comment, while some suggestions I put below:

The article could benefit from a broader discussion on the implications of these findings for the management of trunk canker disease in Chinese hickory and other susceptible crops. More discussion will help readers to catch the important of results in this study.

While the identification of gene families related to pathogenicity is valuable, a deeper analysis of their functional roles and interaction with the host's defense mechanisms could provide more insights. Incorporating bioinformatics tools or experimental validation to predict or confirm the function of these genes would strengthen the findings. More discussion will help to solve this point.

The manuscript would benefit from a more thorough discussion of its limitations. Addressing potential biases in the genomic analysis, the representativeness of the gene families identified, and the limitations of comparative genomics in predicting functional roles could provide a more balanced view of the research.

Expanding the comparative analysis to include more species within and outside the Botryosphaeriaceae family could provide a broader evolutionary context. This would help in understanding if the identified gene families are truly unique or expanded in B. dothidea or if similar patterns are observed in other pathogens.

Although the overall language quality is good, a thorough proofreading by a native English speaker could help to polish the manuscript and fix subtle grammatical or stylistic issues that might detract from the professional quality of the writing.

The research manuscript discusses the comparative genomics of Botryosphaeriaceae, focusing on the gene families in Botryosphaeria dothidea related to its pathogenicity on Chinese hickory trees. The study presents a detailed analysis of high-quality genome assemblies, revealing syntenic relationships between B. dothidea and related species and the conservative evolution within the Botryosphaeriaceae family. It identifies specific genes in B. dothidea that are either unique or expanded, suggesting their role in the pathogen's ability to infect Chinese hickory trees. These include genes encoding for viral integrases, heterokaryon incompatibility proteins, secreted chitinases, and enzymes for orsellinic acid biosynthesis. The aims of this study is clear and the results in interest to me. I don’t have major comment, while some suggestions I put below:

The article could benefit from a broader discussion on the implications of these findings for the management of trunk canker disease in Chinese hickory and other susceptible crops. More discussion will help readers to catch the important of results in this study.

While the identification of gene families related to pathogenicity is valuable, a deeper analysis of their functional roles and interaction with the host's defense mechanisms could provide more insights. Incorporating bioinformatics tools or experimental validation to predict or confirm the function of these genes would strengthen the findings. More discussion will help to solve this point.

The manuscript would benefit from a more thorough discussion of its limitations. Addressing potential biases in the genomic analysis, the representativeness of the gene families identified, and the limitations of comparative genomics in predicting functional roles could provide a more balanced view of the research.

Expanding the comparative analysis to include more species within and outside the Botryosphaeriaceae family could provide a broader evolutionary context. This would help in understanding if the identified gene families are truly unique or expanded in B. dothidea or if similar patterns are observed in other pathogens.

Although the overall language quality is good, a thorough proofreading by a native English speaker could help to polish the manuscript and fix subtle grammatical or stylistic issues that might detract from the professional quality of the writing.

Author Response

Reply to Reviewer 1

The research manuscript discusses the comparative genomics of Botryosphaeriaceae, focusing on the gene families in Botryosphaeria dothidea related to its pathogenicity on Chinese hickory trees. The study presents a detailed analysis of high-quality genome assemblies, revealing syntenic relationships between B. dothidea and related species and the conservative evolution within the Botryosphaeriaceae family. It identifies specific genes in B. dothidea that are either unique or expanded, suggesting their role in the pathogen's ability to infect Chinese hickory trees. These include genes encoding for viral integrases, heterokaryon incompatibility proteins, secreted chitinases, and enzymes for orsellinic acid biosynthesis. The aims of this study is clear and the results in interest to me. I don’t have major comment, while some suggestions I put below:

The article could benefit from a broader discussion on the implications of these findings for the management of trunk canker disease in Chinese hickory and other susceptible crops. More discussion will help readers to catch the important of results in this study.

Reply: Thanks for your kindly advices! We added the discussion about: investigation of B. dothidea’s pathogenicity-related genes could benefits for the management of trunk canker disease control, which provided targeted genes used for fungicides development in future. You can find it at line 85-94 of revised manuscript, so that provide a clear reasoning line for readers as soon as possible: 1) B. dothidea is dominant pathogen; 2) Chinese hickory control strategy face issue of low efficiency; 3) exploring B. dothidea’s pathogenicity-related genes is necessary for development of more efficient fungicides in future. Thus, the readers can get implications of these findings at introduction part.

While the identification of gene families related to pathogenicity is valuable, a deeper analysis of their functional roles and interaction with the host's defense mechanisms could provide more insights. Incorporating bioinformatics tools or experimental validation to predict or confirm the function of these genes would strengthen the findings. More discussion will help to solve this point.

Reply: Thanks for your kindly advices! In newest revised manuscript, we selected one secreted glucanase, two secreted chitinase, two orsellinic acid biosynthesis enzymes, three major facilitator superfamily proteins, one viral integrase and one LRR-containing protein significantly upregulated during infection stage. Among them, one secreted glucanase, one orsellinic acid biosynthesis enzyme and two MFS transporters were verified that involve in B. dothidea’s pathogenicity. You can find it at line 413-439,527-531,540-544,561-563,570-574 of revised manuscript.

The manuscript would benefit from a more thorough discussion of its limitations. Addressing potential biases in the genomic analysis, the representativeness of the gene families identified, and the limitations of comparative genomics in predicting functional roles could provide a more balanced view of the research.

Reply: Thanks for your kindly advices! We have added limitations of this comparative analysis at line 577-582 of revised manuscript.

Expanding the comparative analysis to include more species within and outside the Botryosphaeriaceae family could provide a broader evolutionary context. This would help in understanding if the identified gene families are truly unique or expanded in B. dothidea or if similar patterns are observed in other pathogens.

Reply: Thanks for your kindly advices! The reason we only employed species from Botryosphaeriaceae family is: B. dothidea is dominant pathogen among pathogens isolated from trunk canker diseased samples, which mainly include species from Botryosphaeriaceae family (our previous research at Ref 23). Thus, our purpose is exploring genome differences between B. dothidea and its closed species in Botryosphaeriaceae family, which may help to explain why B. dothidea is dominant in trunk canker diseased samples. However, if we used species outside the Botryosphaeriaceae family, the differences between Botryosphaeriaceae family and outside species will cover the differences between B. dothidea and other species from Botryosphaeriaceae family.

Although the overall language quality is good, a thorough proofreading by a native English speaker could help to polish the manuscript and fix subtle grammatical or stylistic issues that might detract from the professional quality of the writing.

Reply: Thanks for your kindly advices! We have fixed grammatical or stylistic issues in revised manuscript.

Reviewer 2 Report

This paper concerns genomic comparisons between Botryosphaeria dothidea, a parasitic fungus of Chinese hickory, and closely related species. Although this study has a large amount of data and there are no particular problems with the content of the study or its analytical methods, the findings from this study do not appear to contribute to the elucidation of the pathogenicity of B. dothidea against Chinese hickory. It may be the next step. Rather, I evaluated this study as a base-line about pathogenicity of B. dothidea.

The authors specifically aim to investigate virulence genes involved in infection as well as to search for genes that may be advantageous within the fungal community within trunk canker.

The authors wondered if the dominant role of B. dothidea in stem canker disease of Chinese hickory might be related to stronger colony-forming ability or virulence. This study explores clues from a genomic perspective as to the factors that contribute to B. dothidea's dominant role in hickory trunk canker disease.

This study explored candidate genes associated with pathogenicity of B. dothidea through genomic comparisons of eight Botryosphaeriaceae species. We also identified genes encoding enzymes similar to those reported in existing reports, such as chitinases. The related genes to viral integration and heterokaryon incompatibility also .

There is no problem on the methodology.

The authors specifically aim to investigate virulence genes involved in infection as well as to search for genes that may be advantageous within the fungal community within trunk canker. By comparing genomes among Botryosphaeriaceae, the authors have identified chitinase genes and orcellinic acid biosynthesis-related genes that may be advantageous within the community, in addition to pathogenicity-related genes, although the function of the genes were already shown in previous other reports.

My specific suggestion is only on the text format. Line spacing of the line 33-76, 85-364, 414-480, all Tables should be narrowed. Also, location of figures should be close to the related texts.

Author Response

Reply to Reviewer 2

Major comments

This paper concerns genomic comparisons between Botryosphaeria dothidea, a parasitic fungus of Chinese hickory, and closely related species. Although this study has a large amount of data and there are no particular problems with the content of the study or its analytical methods, the findings from this study do not appear to contribute to the elucidation of the pathogenicity of B. dothidea against Chinese hickory. It may be the next step. Rather, I evaluated this study as a base-line about pathogenicity of B. dothidea.

Reply: Thanks for your kindly advices! In revised manuscript, we find one secreted glucanase, two secreted chitinase, two orsellinic acid biosynthesis enzymes, three major facilitator superfamily proteins, one viral integrase and one LRR-containing protein significantly upregulated during infection stage. Among them, one secreted glucanase, one orsellinic acid biosynthesis enzyme and two MFS transporters were verified that involve in B. dothidea’s pathogenicity. You can find it at line 413-439,527-531,540-544,561-563,570-574 of revised manuscript.

Detail comments

The authors specifically aim to investigate virulence genes involved in infection as well as to search for genes that may be advantageous within the fungal community within trunk canker.

Reply: Thanks for your kindly comments!

The authors wondered if the dominant role of B. dothidea in stem canker disease of Chinese hickory might be related to stronger colony-forming ability or virulence. This study explores clues from a genomic perspective as to the factors that contribute to B. dothidea's dominant role in hickory trunk canker disease.

Reply: Thanks for your kindly comments!

This study explored candidate genes associated with pathogenicity of B. dothidea through genomic comparisons of eight Botryosphaeriaceae species. We also identified genes encoding enzymes similar to those reported in existing reports, such as chitinases. The related genes to viral integration and heterokaryon incompatibility also.

Reply: Thanks for your comments! We also verified MFS transporters and orsellinic acid biosynthesis enzyme are associated with B. dothidea’s pathogenicity as well. Hope it will help to design corresponding fungicide in further researches.

There is no problem on the methodology.

The authors specifically aim to investigate virulence genes involved in infection as well as to search for genes that may be advantageous within the fungal community within trunk canker. By comparing genomes among Botryosphaeriaceae, the authors have identified chitinase genes and orcellinic acid biosynthesis-related genes that may be advantageous within the community, in addition to pathogenicity-related genes, although the function of the genes were already shown in previous other reports.

Reply: Thanks for your comments! Beside the chitinase and orcellinic acid biosynthesis-related genes, the relationship between MFS transporters and B. dothidea’s pathogenicity has been also verified in the revised manuscript.

My specific suggestion is only on the text format. Line spacing of the line 33-76, 85-364, 414-480, all Tables should be narrowed. Also, location of figures should be close to the related texts.

Reply: Thanks for your kindly advices! We have narrowed all Tables in this manuscript, and enlarged the font size as well. We also made the caption closer to corresponding figures.

Reviewer 3 Report

The manuscript is fine. I do not have many comments but I think it does not advance the cause it aimed to advance enough. The finding about the LS regions are interesting but it is not clear how relevant are they to pathogenesis. The positive selected LRR domain might be relevant but it was not further tested. Bottom line I disagree with the abstract statement "his study contributes to our understanding of B. dothidea’s pathogenicity and potential influence on the infection of Chinese hickory"

At least perform expression analysis of suspected genes during infection if not a genetic analysis

Author Response

Reviewer 3

Major comments

The manuscript is fine. I do not have many comments but I think it does not advance the cause it aimed to advance enough. The finding about the LS regions are interesting but it is not clear how relevant are they to pathogenesis. The positive selected LRR domain might be relevant but it was not further tested. Bottom line I disagree with the abstract statement "his study contributes to our understanding of B. dothidea’s pathogenicity and potential influence on the infection of Chinese hickory"

Reply: Thanks for your kindly advices! In revised manuscript, we find one secreted glucanase, two secreted chitinase, two orsellinic acid biosynthesis enzymes, three major facilitator superfamily proteins, one viral integrase and one LRR-containing protein significantly upregulated during infection stage. Among them, one secreted glucanase, one orsellinic acid biosynthesis enzyme and two MFS transporters were verified that involve in B. dothidea’s pathogenicity. You can find it at Fig 6-7 and line 413-439,527-531,540-544,561-563,570-574 of revised manuscript. These results provide evidence to support ‘contributes to our understanding of B. dothidea’s pathogenicity’. Meanwhile, BDLA_00007003 (orsellinic acid biosynthesis enzyme) and BDLA_00010781 (MFS transporter) also locate in LS region.

Detail comments

At least perform expression analysis of suspected genes during infection if not a genetic analysis

Reply: we have performed qRT-PCR assay for one secreted glucanase, two secreted chitinase, two orsellinic acid biosynthesis enzymes, three major facilitator superfamily proteins, one viral integrase and one LRR-containing protein, which belong to gene family investigated by bioinformatics analysis. You can find it at Fig 6.

Round 2

Reviewer 3 Report

No further comments

I do not have comments

Author Response

Thanks for your patient and kindly advices!